# Evaluation of Vertebrate-Specific Replication-Defective Zika Virus, a Novel Single-Cycle Arbovirus Vaccine, in a Mouse Model

**DOI:** 10.3390/vaccines9040338

**Published:** 2021-04-01

**Authors:** Shengfeng Wan, Shengbo Cao, Xugang Wang, Yanfei Zhou, Weidong Yan, Xinbin Gu, Tzyy-Choou Wu, Xiaowu Pang

**Affiliations:** 1State Key Laboratory of Agricultural Microbiology, Huazhong Agricultural University, Wuhan 430070, China; sfwan@webmail.hzau.edu.cn (S.W.); sbcao@mail.hzau.edu.cn (S.C.); wxg13277278849@163.com (X.W.); yanwd2007@163.com (W.Y.); 2Department of Oral Pathology, College of Dentistry, Howard University, Washington, DC 20059, USA; xgu@howard.edu; 3Tegen Biomedical Co., Rockville, MD 20851, USA; Yanfei.Zhou@milliporesigma.com; 4Department of Molecular Microbiology & Immunology, Johns Hopkins Medical Institutes, Baltimore, MD 2128, USA; wutc@jhmi.edu

**Keywords:** vertebrate-specific replication-defective Zika virus, single-cycle arbovirus vaccine, artificial insect-specific virus, cellular and humoral immune response, immunity, safety

## Abstract

The flavivirus Zika (ZIKV) has emerged as a global threat, making the development of a ZIKV vaccine a priority. While live-attenuated vaccines are known to induce long-term immunity but reduced safety, inactivated vaccines exhibit a weaker immune response as a trade-off for increased safety margins. To overcome the trade-off between immunogenicity and safety, the concept of a third-generation flavivirus vaccine based on single-cycle flaviviruses has been developed. These third-generation flavivirus vaccines have demonstrated extreme potency with a high level of safety in animal models. However, the production of these single-cycle, encapsidation-defective flaviviruses requires a complicated virion packaging system. Here, we investigated a new single-cycle flavivirus vaccine, a vertebrate-specific replication-defective ZIKV (VSRD-ZIKV), in a mouse model. VSRD-ZIKV replicates to high titers in insect cells but can only initiate a single-round infection in vertebrate cells. During a single round of infection, VSRD-ZIKV can express all the authentic viral antigens in vertebrate hosts. VSRD-ZIKV immunization elicited a robust cellular and humoral immune response that protected against a lethal ZIKV challenge in AG129 mice. Additionally, VSRD-ZIKV-immunized pregnant mice were protected against vertically transferring a lethal ZIKV infection to their offspring. Immunized male mice were protected and prevented viral accumulation in the testes after being challenged with lethal ZIKV. Overall, our results indicate that VSRD-ZIKV induces a potent protective immunity against ZIKV in a mouse model and represents a promising approach to develop novel single-cycle arbovirus vaccines.

## 1. Introduction

The Zika virus (ZIKV), a mosquito-borne flavivirus, was first discovered in 1947 and was considered unimportant until it emerged in the Pacific region and quickly spread to South America, Central America, and the Caribbean in 2015 [1,2,3]. The explosive epidemics and its association with Guillain–Barré syndrome [4], a devastating autoimmune disorder targeting the nervous system, and congenital disabilities, including microcephaly [5], led the World Health Organization to declare ZIKV a “Public Health Emergency of International Concern” in February 2016 [6,7]. Since then, various ZIKV vaccine platforms have been studied to control future epidemics [8]. The platforms that have been or are under study include live virus [9], inactivated virus [10], chimeric virus [11], virus-like particles (VLP) [12], and a subunit vaccine [13] as well as vaccines based upon messenger RNA (mRNA) [14], DNA [15], protein [16], and vector-based formulations [17]. Typically, attenuated viral vaccines provide long-lasting immunity but reduced safety; in contrast, inactivated vaccines provide a high initial safety level but exhibit weak long-term immunity [18,19]. Widman et al. [20] developed a flavivirus vaccine based on single-cycle, encapsidation-defective viruses to avoid sacrificing immunogenicity to provide sufficient safety. However, single-cycle, encapsidation-defective viruses require complicated packaging systems [21] or helper viral replicons [22], making the development of a licensed commercial vaccine difficult. Recently, we converted a dual-host ZIKV into a vertebrate-specific replication-defective ZIKV (VSRD-ZIKV) and demonstrated its inherently high level of safety in immunocompromised mice [23]. Therefore, the VSRD-ZIKV can replicate efficiently in mosquito-derived C6/36 cells and overcome packaging cell dependence to produce encapsidation-defective viruses. In the present study, we investigated the immunogenicity of this new single-cycle ZIKV in mouse models as an attempt to produce a vaccine with a high degree of safety and immunogenicity. In addition, VSRD-ZIKV could be grown to a high titer in insect cells for the ease of production. 

## 2. Materials and Methods

### 2.1. Cell Cultures, Virus Stocks, Escherichia coli, Yeast Strains, and Antibodies

Vero cells (African Green Monkey Kidney Epithelial Cells) were propagated in Dulbecco’s Modified Eagle’s Medium (DMEM) (Life Technologies, Carlsbad, CA, USA) supplemented with 10% fetal bovine serum (FBS) at 37 °C under 5% CO_2_. *Aedes albopictus* C6/36 cells (American Type Culture Collection (ATCC), Manassas, VA, USA, CRL-1660) were cultured in Minimum Essential Media (MEM)(Life Technologies) supplemented with 10% FBS and 1% nonessential amino acids at 28 °C in a 5% CO_2_ incubator. Zika virus (MR766 strain) was purchased from ATCC. VSRD-ZIKV was derived from ZIKV/MR766 by a combination of reverse genetics and host-specific adaptation [23]. Dr. Qiyi Tang kindly provided ZIKV from Puerto Rico (PRVABC59). Competent *E. coli* strain Stbl4^TM^ was obtained in the frozen state from Thermo Fisher Scientific (Waltham, MA, USA). *Saccharomyces cerevisiae* YPH857 and mouse monoclonal antibody (mAb) 4G2 cross-reactive vs. flavivirus E protein were purchased from the ATTC. The ZIKV NS5-specific murine monoclonal antibody was produced by our laboratory. Goat anti-mouse Immunoglobulin G (IgG) conjugated with Alexa Fluor 488 was acquired from Sigma. 

### 2.2. Plaque Assay 

Serial 10-fold dilutions of the virus were used to infect Vero or C6/36 cell monolayers in 12-well plates. Cells were incubated with viruses for 1 h at 37 °C (for Vero cells) or 28 °C (for C6/36 cells). The cells were washed with serum-free DMEM and further incubated for 3 to 5 days in DMEM containing 3% fetal bovine serum and 2% sodium carboxymethyl cellulose (SCMC; Sigma, Munich, Germany). After removing the SCMC, cells were fixed with 4% formaldehyde and incubated at room temperature for 2 h. The fixative was removed, and the cell monolayer were stained with 1% crystal violet for 1 h. Visible plaques were counted, and the viral titers were calculated. All data were expressed as the mean of triplicate samples.

### 2.3. Fluorescent Focus Assay and Immunostaining

We performed a fluorescent focus assay using serial tenfold dilutions (10^1^–10^6^) of each sample in DMEM. Two hundred microliters (200 µL) of each dilution was added to 90% confluent Vero cells.in a 24-well plate. After 1 h of incubation at 37 °C, the supernatant was discarded. Each well was added with 500 μL of methylcellulose overlay containing 2% FBS and 1% penicillin/streptomycin. After 24 h of incubation at 37 °C, the methylcellulose overlay was removed, and the cells were fixed in acetone at −20 °C for 15 min. The fixative was removed, and the plates were air-dried and washed three times with Phosphate-buffered saline (PBS) before being incubated in PBS with 1% FBS for 1 h. Fixed cells were reacted with a mouse monoclonal antibody 4G2 or NS5 monoclonal antibody for 1 h. Afterwards, plates were washed three times with PBS and incubated with Alexa Fluor 488 goat anti-mouse IgG for 1 h in the blocking buffer; after which, the cells were washed three times with PBS. Samples were observed, and the positive cell number was calculated with a Leitz fluorescent microscope.

### 2.4. VSRD-ZIKV Vaccination and Challenge on AG129 Mice

The Huazhong Agricultural University IACUC approved all animal experiments (Ethic code: HZAUMO-2016-035). AG129 mice maintained under specific pathogen-free conditions were obtained from the Wuhan Institute of Virology. Two groups of 5-week-old AG129 mice (*n* = 9) were immunized with 1 × 10^6^ FFU (Fluorescent Focus Units) VSRD-ZIKV by the subcutaneous (s.c.) route. Controls received DMEM subcutaneously. On day 14 post-immunization (p.i.), one of the two groups was immunized with 1 × 10^6^ FFU VSRD-ZIKV again. At 28 days p.i., mice were tested for the antibody using a neutralization assay. Immunized mice were then challenged through the intraperitoneal (i.p.) route with 1 × 10^3^ FFU of the ZIKV PRVABC59 strain; disease signs were assessed daily for 21 days, and viremia was measured on days 2 and 4 after the challenge. The serum was collected by centrifuging the blood at 3000× *g* for 5 min at romm temperature and stored at −80 °C. Viral titers were determined by an immunostaining focus assay on Vero cells, as described above.

### 2.5. Intracellular Cytokine Staining (ICS)

Four groups of 5-weeks-old A129 mice (*n* = 4) were immunized with DMEM, 1 × 10^5^ FFU VSRD-ZIKV, 1 × 10^5^ FFU ZIKV MR766 WT, or 1 × 10^5^ FFU ZIKV PRVABC59 through the s.c. route. On day seven p.i., all animals were sacrificed, and the spleens were isolated to perform the ICS experiment. Approximately 2 × 10^6^ splenocytes were stimulated with E polypeptide array (strain PRVABC59) (Bioresource) overnight. At the same time, BD GolgiPlug (BD Bioscience, San Jose, CA, USA) was added to block protein transport. Cells were incubated with antibodies to CD4 or CD8, then fixed with 2% paraformaldehyde. Cells were permeabilized before the addition of anti-interferon-γ (IFN-γ) or control IgG1 (rats) (e-Biosciences). All samples were processed with a C6 Flow Cytometer instrument (BD Biosciences). Dead cells were excluded based on the forward and side scatter. Data were analyzed with FlowJo.

### 2.6. IFN-γ Immunoassay

Approximately 4 × 10^5^ splenocytes were plated in 12-well plates and stimulated with E polypeptide (strain PRVABC59) for 48 h. Culture supernatants were harvested, and IFN-γ production was measured using a Mouse Cytokine Assay (Invitrogen, Carlsbad, CA, USA).

### 2.7. RNA Extraction and Quantitative Real-Time (RT)-PCR

We extracted total RNA using a Trizol reagent (Invitrogen. One microgram of RNA was used to synthesize complementary DNA (cDNA) using the first-strand cDNA kit from the SuperScript III System (Thermo Fisher Scientific, Waltham, MA, USA). Quantitative real-time (RT)-PCR was performed using the Applied Biosystems 7500 and SYBR green PCR master mix (Toyobo). We normalized the data based upon the level of β-actin expression in each sample. The primers utilized are listed in Appendix A. The relative expression of Tumor necrosis factor (TNF-α), chemokine ligand 5 (CCL-5), Interleukin 1 beta (IL-1β), and Interleukin 6 (IL-6) were normalized to the levels of endogenous control β-actin within each sample using the 2−ΔΔCT (where CT is the threshold cycle) method. Copies of viral RNA in samples were calculated with calibration samples using the 2−ΔΔCT method.

### 2.8. Haematoxylin and Eosin (H&E) Staining

Tissues were obtained from AG129 mice after they were anesthetized with ketamine-xylazine (0.1 mL per 10 g of body weight). Tissues were embedded in paraffin for coronal sections. The resulting sections were H&E stained as follows. Tissues were fixed for 48 h at 4 °C to obtain optimal tissue integrity. Serial sections were mounted on polysilinated microscope glass slides (Thermo Fisher Scientific, Waltham, MA, USA) and dried at room temperature overnight. Tissue samples were stained for 2.5 min in hematoxylin, 15 s in 1% acetic industrial methylated spirits, and 15 s in ammoniated water, and 4 min in eosin. The PAS (Periodic Acid Schiff) stain included immersion in a periodic acid–Schiff solution for 15 min, followed by staining with haematoxylin for 3 min. After staining, sections were dehydrated using an ethanol series. Microscopic images were captured using a Leica CTR500 microscope camera with a bright field light source (Leica Microsystems Inc., Wetzlar, Germany)

### 2.9. Statistical Analysis

Experiments were performed three or more times under similar conditions. Analyses were conducted using GraphPad Prism, version 5 (GraphPad Software, San Diego, CA, USA). Nearly all results were expressed as the mean ± standard error (SEM) The only exception was that the viral loads were expressed as the median. Statistical differences between the experimental groups were established using a two-way analysis of variance (ANOVA) and *t*-tests using the Bonferroni post-hoc test for multiple comparisons. A *p*-value of < 0.05 was considered significant.

## 3. Results

### 3.1. Prime-Boost VSRD-ZIKV Vaccination Provides Robust Protection Against Lethal ZIKV Challenge in Immunocompromised Mice 

In our previous study, a high-dose challenge with VSRD-ZIKV was safe in newborn and three-week-old immunocompromised mice [23]. Given the high level of safety exhibited by VSRD-ZIKV, we next assessed the protective efficacy of a single delivery and prime-boost VSRD-ZIKV vaccination regimen in five-week-old AG129 mice. Two groups of mice (*n* = 7/group for single immunization and *n* = 9/group for primary-boost immunization) were immunized with 10^6^ FFU VSRD-ZIKV via the s.c. route. Two weeks post-immunization, the prime-boost group received a second immunization with 10^6^ FFU VSRD-ZIKV. The sham group received an s.c. injection of 100-μL DMEM. None of the mice exhibited any visible differences in mobility, behavior, body weight, or temperature following the immunization. Two weeks following the second immunization, serum was collected, and the mice were challenged with 10^3^ PFU (Plague Forming Units) of the ZIKV PRVABC59 strain via the i.p. route. All vaccinated mice remained healthy throughout the experiment, whereas the mice from the sham group began to show signs of sickness on Day 8, and all had died by 13 days post-ZIKV challenge (Figure 1A–C). We then assessed the serum viral load in the challenged mice by FFU. In contrast to the sham immunization, a single-dose immunization with VSRD-ZIKV significantly reduced the serum viral load. The prime-boost group displayed the undetectable infectious virus on Day 2 and Day 4 post-infection (Figure 1D,E). 

Given the robust protection elicited following the ZIKV challenge in AG129 mice, we next wondered whether VSRD-ZIKV would protect wild-type mice following a challenge with a more pathogenic strain of wild-type ZIKV. Due to the general resistance of immunocompetent strains to ZIKV infection and disease [24,25], we used A129 mice, which contain a functional IFN-γ receptor but lacked an IFN-α/β receptor challenged with the French Polynesian H/PF/2013 ZIKV isolate as a model. Using the same experimental design, five-week-old A129 mice were immunized once or immunized and subsequently boosted with VSRD-ZIKV before the lethal challenge with 4 × 10^2^ PFU H/PF/2013 ZIKV (the median lethal dose, LD_50,_ is 8 PFU in A129 mice). Similar to the results of the AG129 mice, the sham vaccinated mice began to show signs of illness on Day 8 and succumbed to the infection by Day 14 (Figure 1F–H). In contrast, both the groups that received either a single dose or the prime-boost regimen remained healthy until the end of the experimental period. The viral titer in the sham-vaccinated mice’s serum was two to three logs higher than that of the VSRD-ZIKV-vaccinated groups (Figure 1I). 

### 3.2. Vaccination of VSRD-ZIKV is Associated with ZIKV-Specific Humoral and Cellular Immune Responses

Given the robust protection against the lethal ZIKV challenge afforded by the immunization with VSRD-ZIKV, we sought to characterize the humoral and T-cell responses generated by VSRD-ZIKV immunization. The prime-boost scheme’s higher protective efficacy was supported by a higher serum neutralizing antibody titer (Figure 2A). Additionally, the production of E-, NS1-, or NS5-specific IgG demonstrated that the VSRD-ZIKV-based vaccine induced an immune response against both structural and nonstructural viral proteins, supporting the expression of the entire viral genome through a single-round infection of VSRD-ZIKV in the host (Figure 2B–D).

Typically, robust T-cell immunity is associated with live virus vaccines [26]. Moreover, T-cell immunity has been shown to play an essential role in preventing ZIKV infection [27]. Due to the general resistance of immunocompetent mice to ZIKV infection and disease [28], we evaluated the T-cell responses in an A129 mouse model deficient in interferon α/β receptors. Groups of six-week-old female A129 mice (*n* = 4/group) were immunized with 10^5^ FFU of VSRD-ZIKV or 10^5^ FFU of WT ZIKV-MR766 or WT ZIKA-PRVABC59 via the s.c. route. The control group received an s.c. injection of 100-μL DMEM. On Day 7 post-immunization, ZIKV-specific T cells isolated from the spleen were stimulated in vitro with an E peptide array and analyzed using intracellular cytokine staining (ICS) and flow cytometry analysis. Both WT ZIKV and VSRD-ZIKV generated a comparable ZIKV-specific CD4^+^ T-cell response. In contrast, the VSRD-ZIKV-immunized animals displayed significantly greater ZIKV-specific CD8^+^ T cells than WT ZIKV-infected animals (Figure 2E). Furthermore, the total number of T cells from the VSRD-ZIKV-immunized mice produced more IFN-γ than the WT ZIKV-infected mice (Figure 2F). 

### 3.3. VSRD-ZIKV Provides Protection and Prevented Virus Accumulation in the Testes of Male Mice Challenged with Lethal ZIKV

Since ZIKV accumulation in the testes and sperm of males is a concern for viral transmission [29], we next aimed to determine if a vaccination with VSRD-ZIKV could confer protection in the testes of immunocompromised male mice against the lethal ZIKV challenge. To this end, five-week-old male AG129 mice (*n* = 4) were immunized with 1 × 10^6^ FFU VSRD-ZIKV or DMEM through the s.c. route. On day 14 p.i., the 1 × 10^6^ FFU VSRD-ZIKV group was immunized with 1 × 10^6^ FFU VSRD-ZIKV again. At 28 days p.i., the immunized mice were challenged with 1 × 10^3^ PFU of the ZIKV PRVABC59 strain via the i.p. route. Seven days later, the mice were euthanized and then necropsied. Immediately after the epididymis and testes were removed, the number of motile sperm and nonmotile sperm were counted on a hemocytometer. The viral loads in the heart, liver, spleen, lung, kidney, brain, and testes were measured by qRT-PCR. The viral RNA level was lower in the vaccinated mice than the sham group, with level two to four logs higher in the sham group (Figure 3A). A histological analysis of the testes revealed that the sham group displayed enhanced inflammatory cell infiltration and damage, which was substantially reduced in the VSRD-ZIKV-immunized group’s testes (Figure 3B). The viral RNA level in the sperm of the mice immunized with VSRD-ZIKV was barely above the detection level, whereas the viral load was significantly higher in the sham-challenged mice (*p* < 0.001; Figure 3C). Moreover, both the total number of sperm and the number of motile sperm in the challenged-VSRD-ZIKV mice were comparable to the unchallenged control mice. In contrast, the sham-challenged mice were significantly impaired (Figure 3D,E). Collectively, these findings indicate that immunization with VSRD-ZIKV protected the male testes and sperm from lethal ZIKV infection. 

### 3.4. Immunization with VSRD-ZIKV Protects against Vertical Transmission in Pregnant Mice Challenged with Lethal ZIKV

Zika virus infection during pregnancy is a cause of microcephaly and other congenital abnormalities in the developing fetus and newborns [18,19]. Therefore, we examined whether an immunization with VSRD-ZIKV could protect pregnant mice from transferring a lethal ZIKV infection to their offspring. Female AG129 mice that received a prime-boost regimen of VSRD-ZIKV were mated on day 22 post-immunization. The pregnant mice were then challenged with a lethal ZIKV infection (PRAVABC59 strain). The placenta, fetal brain, and the pregnant mouse brain were assessed 12 days post-infection for cytokine expression and histopathology (Figure 4A). Upon gross examination, the sham control and both the VSRD-ZIKV-immunized control and challenged mice exhibited normal-sized embryos. In contrast, the embryos in the unimmunized challenged mice were substantially reduced in size (Figure 4B). Corresponding with the gross embryo size, the viral loads in the pregnant mouse brain, placenta, and fetal mouse brain were significantly higher in the unimmunized challenged mice than the sham control VSRD-ZIKV-immunized groups, in which there was no detectable virus (Figure 4C). Such a viral burden was also associated with significantly higher levels of proinflammatory cytokines and chemokines in the placenta of the unimmunized-challenged mice compared with the other groups (Appendix A). In the pregnant mouse brain, the level of TNF-α, IL-1β, and CCL5 were elevated in the unimmunized challenged mice. This corresponded with the increased inflammatory cell infiltration, hemorrhaging, and cellular damage observed in the histopathological sections compared to the VSRD-ZIKV-immunized groups and the unchallenged control (Figure 4D). Similar findings were observed in the fetal mouse brain (Figure 4E). Finally, immunohistochemistry was performed on the pregnant mouse brains, revealing that the astrocytes and microglia were activated in only the unimmunized-challenged group (Figure 4F). 

## 4. Discussion

The development of vaccines for vector-borne virus infections is a top public health priority due to the significant global disease burden [20,21]. Currently, there are only three arbovirus vaccines available for the general population, including live attenuated yellow fever virus (YFV) vaccine, inactivated Japanese encephalitis virus (JEV) vaccine, and inactivated tick-borne encephalitis virus (TBEV) vaccine [30]. In response to the explosive ZIKV epidemic in the Americas, several vaccine platforms have been investigated, including inactivated virus [17,31], live attenuated virus [32], DNA vaccine [17,31], and RNA vaccines [26,27]. Nonetheless, a licensed ZIKV vaccine has yet to be produced. VSRD-ZIKV can initiate a typical infection and express all viral antigens in vaccinated mice, similar to a live virus vaccine. On the other hand, the VSRD-ZIKV-based vaccine does not produce infectious virions in a vertebrate host, which may yield a safety level usually observed with inactivated vaccines. In our previous study, we selected AG129 mice as a highly sensitive lethal model to investigate the safety of VSRD-ZIKA. AG129 mice succumb to infection with a dose as low as 1 IFU of wild-type ZIKV [11,28]. The high dose of VSRD-ZIKV used to challenge both newborn and three-week-old immunocompromised mice demonstrated that the VSRD-ZIKV-based vaccine has a high level of safety, similar to inactivated or subunit vaccine high levels of safety and similar to inactivated or subunit vaccines. The vaccination of immunocompromised mice provides robust protection against the lethal WT ZIKV challenge (Figure 1).

Interestingly, the VSRD-ZIKV-immunized animals displayed significantly greater ZIKV-specific CD8+ T cells than the WT ZIKV-infected animals (Figure 2). Although the mechanism by which VSRD-ZIKV immunization enhanced the ZIKV-specific CD8+ T-cell response remains unclear, it may be induced by a higher accumulation of viral antigens. Alternatively, the sorting of viral proteins to the antigen-presenting pathway may differ in the VSRD-ZIKV-infected cells, since they did not release infectious viral particles [33]. The single-round infection of VSRD-ZIKV at the inoculation site where the expressed viral proteins induced a comprehensive immune response similar to that of the live virus when compared to the weaker response seen with an inactivated or subunit vaccine. Taken together, our results demonstrate that VSRD-ZIKV is capable of inducing robust cellular and humoral immunity with a high level of safety.

Significantly, high-titer VSRD-ZIKV can be produced in insect cell lines, which will enable large-scale and cost-effective vaccine production. Importantly, immunization with VSRD-ZIKV protected both the male and pregnant mice from the lethal ZIKV infection (Figure 3) and inhibited the vertical transmission of the virus to the fetus (Figure 4). The histopathological results confirmed the high level of neuroprotection afforded by the VSRD-ZIKV vaccine. 

## 5. Conclusions

VSRD-ZIKV was found to elicit a potent and efficacious immune response in the animal models with a similar spectrum of innate and specific immune responses as a live vaccine. Nonetheless, future studies should dissect whether protection and long-term immune memory are afforded by the humoral or CD8+ T-cell response generated following vaccination. Significantly, the methods described in our present study can potentially be applied to other vector-borne viruses, leading to the development of safe, effective, and affordable vaccines.

## Figures and Tables

**Figure 1 vaccines-09-00338-f001:**
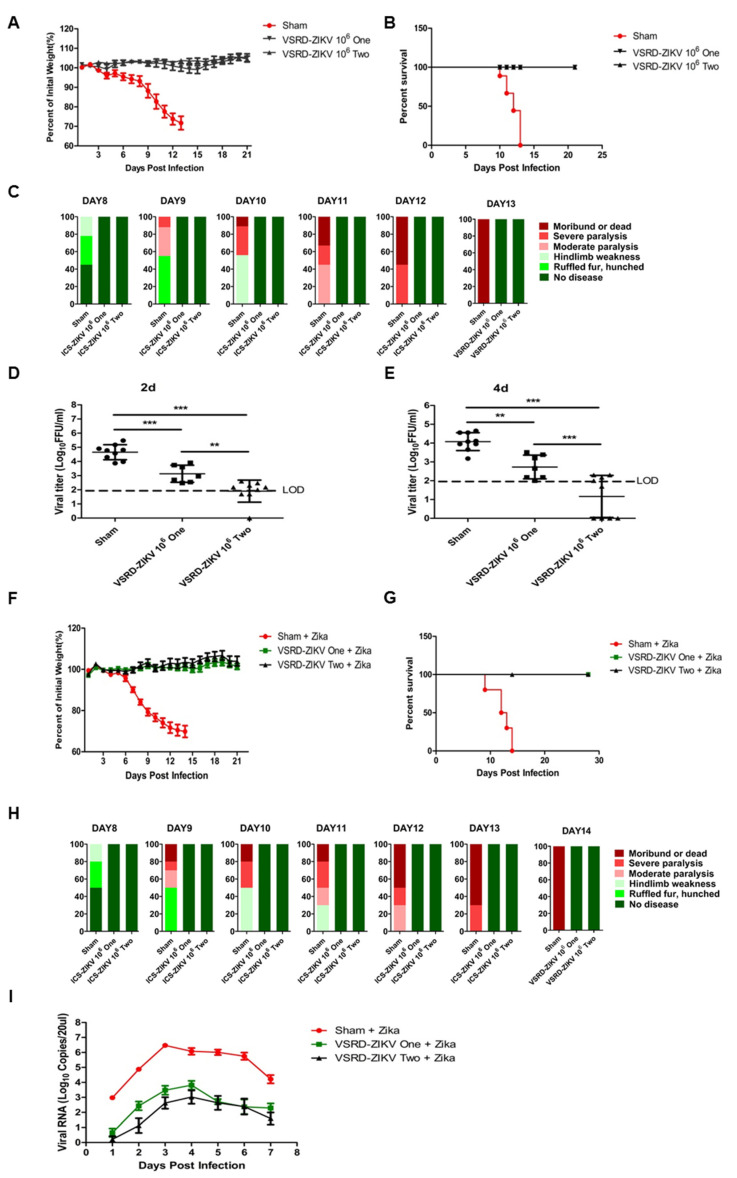
Immunization with the vertebrate-specific replication-defective Zika virus (VSRD-ZIKV) protects against the lethal ZIKV challenge. AG129 mice were immunized and challenged as described in the Methods section. Following the viral challenge, body weight (**A**), survival (**B**), and symptoms (**C**) were monitored. The serum viral load was determined by qRT-PCR on days 2 and 4 ((**D**,**E**), respectively). Five-week-old A129 mice were immunized once or immunized and subsequently boosted with VSRD-ZIKV before the lethal challenge with 4 × 10^2^ PFU H/PF/2013 ZIKV. Following the viral challenge, body weight (**F**), survival (**G**), and symptoms (**H**) were monitored. The kinetics of the serum viral load was determined by qRT-PCR following lethal ZIKV infection (**I**). “Circle”: sham, “square”: single immunization, “triangle”: prime-boost immunization. Statistical significance: ** *p* < 0.01, *** *p* < 0.001.

**Figure 2 vaccines-09-00338-f002:**
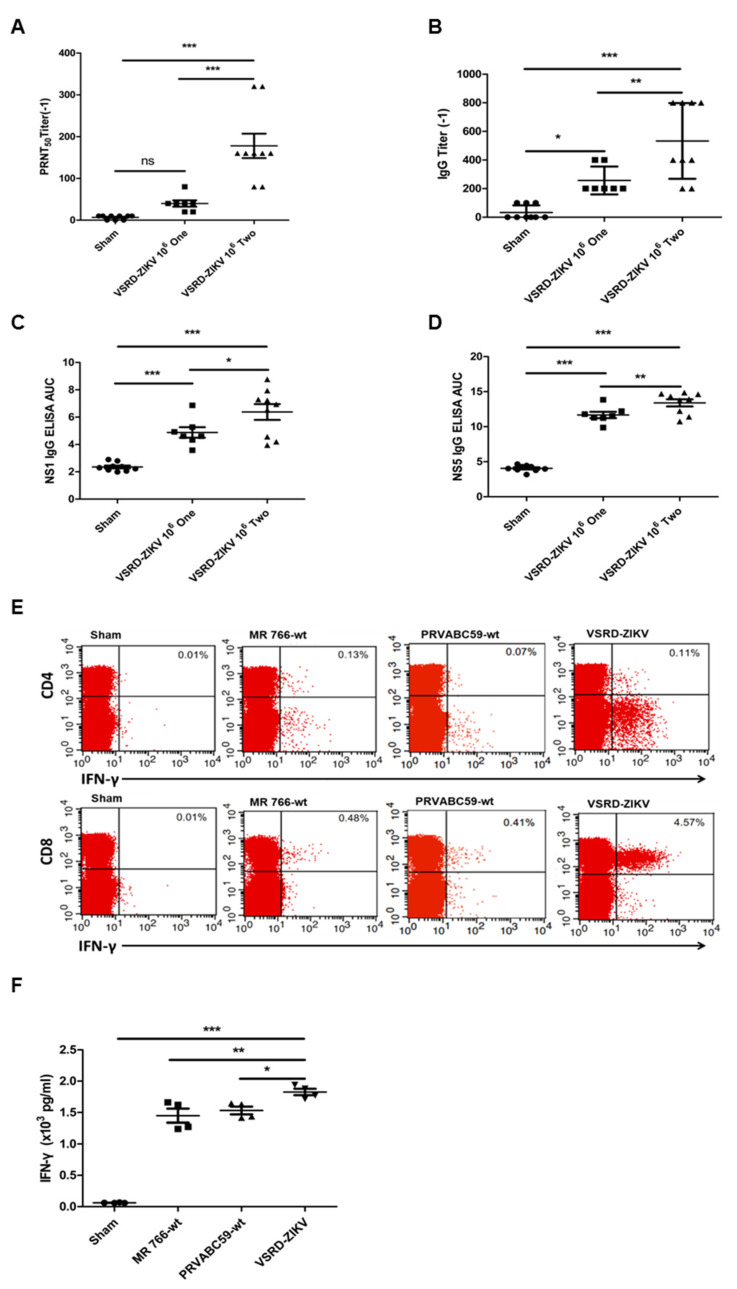
Humoral and cellular immune responses in mice immunized with VSRD-ZIKV. The humoral immune response at 28 days post immunization (p.i.) was determined in immunized AG129 mice: (**A**) the neutralizing titer; (**B**) E-specific antibodies, (**C**) NS1-specific antibodies, and (**D**) NS5-specific antibodies. The cellular immune response was assessed at 7 days p.i.in immunized A129 mice: (**E**) percentages of CD4 +interferon γ (IFN-γ)+ cells and CD8 + IFN-γ+ cells and (**F**) IFN-γ production. (**A**–**D**) “Circle”: sham, “square”: single immunization, “triangle”: prime-boost immunization. (**F**) “Circle”: sham, “square”: ZIKV MR766 Challenge, “triangle” ZIKV PRVABC59 challenge, “reverse triangle”: VSRD-ZIKV challenge. Statistical significance * *p* < 0.05, ** *p* < 0.01, *** *p* < 0.001.

**Figure 3 vaccines-09-00338-f003:**
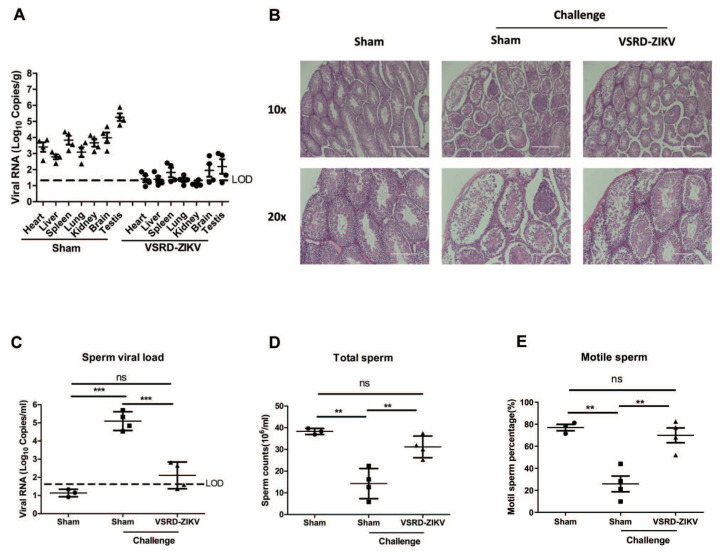
Protective efficacy of VSRD-ZIKV immunization in male mice challenged with lethal ZIKV. Five-week-old male AG129 mice (*n* = 4) were immunized with 1 × 10^6^ FFU VSRD-ZIKV or DMEM through the subcutaneous (s.c.) route and boosted on day 14. On day 28 p.i., the mice were challenged through the intraperitoneal (i.p.) route with 1 × 10^3^ PFU of the ZIKV PRVABC59 strain. Seven days later, the mice were euthanized and necropsied. (**A**) Viral loads in the heart, liver, spleen, lung, kidney, brain, and testes were measured by qRT-PCR. (“triangle”: sham, “circle”: VSRD-ZIKV immunized mice) (**B**) The testes were stained with hematoxylin and eosin (10×: Scale bar = 400 µm 20×: Scale bar = 200 µm). (**C**) The viral load in the sperm was measured by qRT-PCR. The total sperm (**D**) and motile sperm (**E**) were counted using a hemocytometer. (**C**–**E**) “Circle”: sham, “square”: sham challenged with ZIKV, “triangle”: VSRD-ZIKV-immunized mice challenged with ZIKV, Data were expressed as the means ± SEM from three independent experiments. ** *p* < 0.01, and *** *p* < 0.001., ns: Not Significant.

**Figure 4 vaccines-09-00338-f004:**
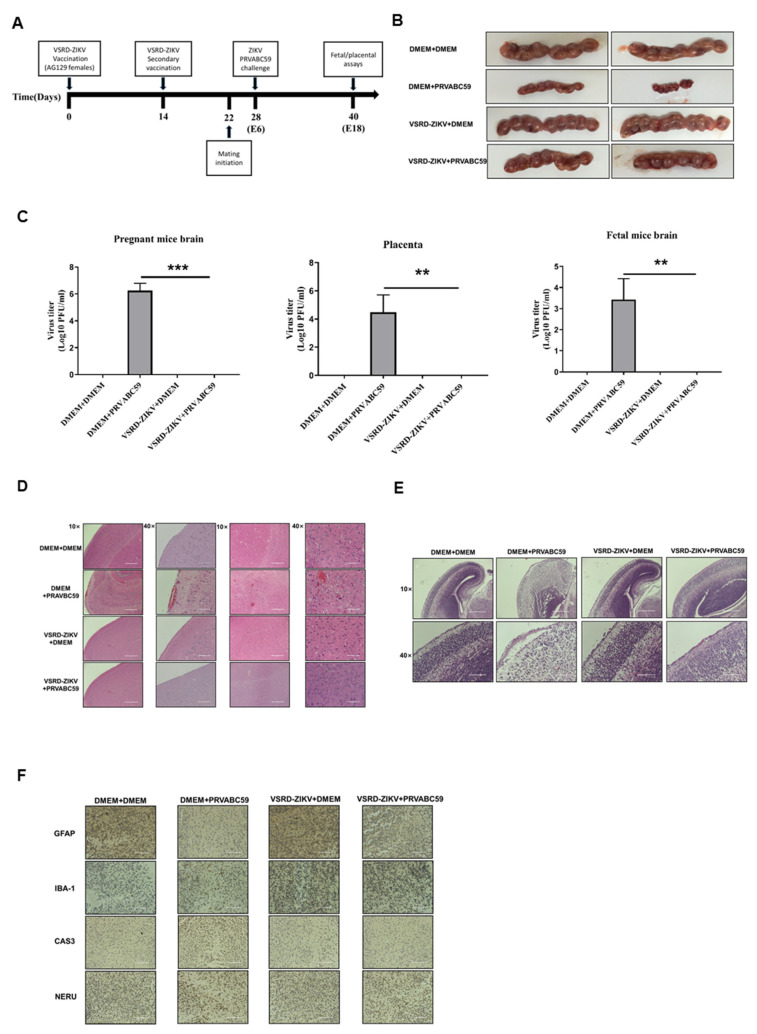
Immunization with VSRD-ZIKV protects against vertical transmission in pregnant mice. (**A**) Experimental schema. Five-week-old female AG129 mice (*n* = 9) were immunized with 1 × 10^6^ FFU VSRD-ZIKV or DMEM through the s.c. route and boosted on day 14. On day 22 p.i., immunized AG129 female mice were mated with naïve male AG129 mice. At E6, mice were inoculated subcutaneously with 10^3^ FFU of ZIKV PRVABC59 via i.p. injection. (**B**) All animals were sacrificed on E18, and the gross pathology of the embryos was assessed. (**C**) The viral loads in the placenta, fetus, and maternal brain tissue were assessed by qRT-PCR. (**D**,**E**) A histological analysis was performed on the placenta, pregnant mouse brain, and fetal brain tissue. (**F**) Immunohistochemistry (IHC) analysis of the brains of pregnant mice. The viral titer was monitored with a plaque assay. Data are expressed as the means ± SEM from three independent experiments. (**D**,**E**) 10×: Scale bar = 400 µm, 40×: Scale bar = 100 µm. (**F**) Scale bar = 100 µm. Statistical significance ** *p* < 0.01, and *** *p* < 0.001.

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
