# Peer review of "Evaluation of Vertebrate-Specific Replication-Defective Zika Virus, a Novel Single-Cycle Arbovirus Vaccine, in a Mouse Model"

_vaccines, 2021, doi:10.3390/vaccines9040338_

Round 1
Reviewer 1 Report
Manuscript titled “Evaluation of vertebrate-specific replication-defective Zika virus, a novel single-cycle arbovirus vaccine, in mouse model” by Shengfeng Wan addresses very important problem with the vaccine development against deadly Zika virus. It is well designed and written paper.
However, several issues have to be addressed:
- Figure 1. It looks like that “clinical” picture of all vaccinated mice was similar regardless the type of vaccination. However, viral loads were different at day 2 and 4 post-infection and some single dose vaccinated mice on day 4 had FFUs similar to unvaccinated animals. What was viral FFU in these animals at days 13 or 15?
- Lines 212 vs. 164: why lower viral concentration has been used in the following experiments? It is difficult to compare the results.
- The authors stated that “In contrast, the VSRD-ZIKV -immunized animals displayed significantly greater ZIKV-specific CD8+ T cells compared to WT ZIKV infected animals (Figure, 218 4E). Furthermore, the total number of T cells from the VSRD-ZIKV -immunized mice produced more IFN-γ than the WT ZIKV-infected mice (Figure, 2F).’” What is the mechanism of such difference?
- Figure 2A: Single dose vaccination produce surprisingly low titer of neutralizing antibodies, which was not statistically different from sham. How the authors explain the success of single does in figure 1?
Why there are two different figure legends for each figure? This is very confusing and very difficult to follow.
All fonts in Figures are of very low resolution, so it is difficult to read and understand some of them.
Manuscript titled “Evaluation of vertebrate-specific replication-defective Zika virus, a novel single-cycle arbovirus vaccine, in mouse model” by Shengfeng Wan addresses very important problem with the vaccine development against deadly Zika virus. It is well designed and written paper.
However, several issues have to be addressed:
- Figure 1. It looks like that “clinical” picture of all vaccinated mice was similar regardless the type of vaccination. However, viral loads were different at day 2 and 4 post-infection and some single dose vaccinated mice on day 4 had FFUs similar to unvaccinated animals. What was viral FFU in these animals at days 13 or 15?
- Lines 212 vs. 164: why lower viral concentration has been used in the following experiments? It is difficult to compare the results.
- The authors stated that “In contrast, the VSRD-ZIKV -immunized animals displayed significantly greater ZIKV-specific CD8+ T cells compared to WT ZIKV infected animals (Figure, 218 4E). Furthermore, the total number of T cells from the VSRD-ZIKV -immunized mice produced more IFN-γ than the WT ZIKV-infected mice (Figure, 2F).’” What is the mechanism of such difference?
- Figure 2A: Single dose vaccination produce surprisingly low titer of neutralizing antibodies, which was not statistically different from sham. How the authors explain the success of single does in figure 1?
Why there are two different figure legends for each figure? This is very confusing and very difficult to follow.
All fonts in Figures are of very low resolution, so it is difficult to read and understand some of them.
Author Response
Thank you for a careful and thorough reading of this manuscript and for thoughtful comments and constructive suggestions, which help improve this manuscript's quality. Our response follows (the reviewer's comments are in italics)
General Comments. Manuscript titled "Evaluation of vertebrate-specific replication-defective Zika virus, a novel single-cycle arbovirus vaccine, in a mouse model" by Shengfeng Wan addresses a very important problem with the vaccine development against deadly Zika virus. It is well designed and written paper.
Response: Thank you very much. We appreciate the positive feedback from you.
However, several issues have to be addressed:
- Figure 1. It looks like that "clinical" picture of all vaccinated mice was similar regardless of the type of vaccination. However, viral loads were different at day 2 and 4 post-infection and some single dose vaccinated mice on day 4 had FFUs similar to unvaccinated animals. What was viral FFU in these animals at days 13 or 15?
Response: Thank you for the thoughtful comment. When we conducted this part of the experiment, both single-dose and prime-boost schemes provided protection of vaccinated mice from lethal challenge. The only difference is that the mice with single dose immunization had higher transient viremia than prime-boost vaccinated mice. For both groups, viremia was not detectable 5 days after immunization.
- Lines 212 vs. 164: why lower viral concentration has been used in the following experiments? It is difficult to compare the results.
Response: One of the study goals is to compare the immunogenicity of the vaccine candidate to natural infection of wild-type virus. We choose a lower challenge dose for this study because WT ZIKVs are highly lethal to both AG129 and A129 mice. If we used the same dose as in the protection experiment (106 FFU WT), the mice would die before the date of collecting the spleen cells (Day 7 post-immunization).
- The authors stated that "In contrast, the VSRD-ZIKV -immunized animals displayed significantly greater ZIKV-specific CD8+ T cells compared to WT ZIKV infected animals (Figure, 218 4E). Furthermore, the total number of T cells from the VSRD-ZIKV -immunized mice produced more IFN-γ than the WT ZIKV-infected mice (Figure, 2F).'” What is the mechanism of such difference?
Response: Thank you for raising this important question. However, the mechanism by which VSRD-ZIKV induced a higher level of ZIKV-specific CD8+ T cells is currently unknown. There may be two possible mechanisms for this observation: 1, Since VSRD-ZIKV-infected mammalian cells do not produce viruses, the viral structural proteins may be accumulated to a higher level in the cells than in WT virus-infected cells. As a result, more viral peptides are presented by MHC class I of VSRD-ZIKV-infected cells; 2, The mechanism for a different level of CD8+ T cell activation may be caused by a different pathway of viral protein degradation and peptide presentation. The enhanced furin cleavage efficiency of viral prM protein results in the formation of matured virions before entering into the Golgi apparatus. When the mature virus passes through the trans-Golgi network (TGN), the low pH environment induces the viral envelope's merging with the Golgi membrane, thereby preventing the virus release from the infected cells. The Golgi membrane-associated viral protein may take different pathways for MHC class I presenting viral peptide, resulting in more effective stimulation of CD8- T cells.
- Figure 2A: Single dose vaccination produce surprisingly low titer of neutralizing antibodies, which was not statistically different from sham. How the authors explain the success of single does in figure 1?
Response: We agree with you that the single-dose vaccination produces a surprisingly lower titer of neutralizing antibodies. Nonetheless, the single-dose vaccination induced a significantly high level of antibodies against non-structural proteins (NS1 and NS5) and a high level of ZIKV-specific CD8+ T cells. Many previous studies have demonstrated that both NS1-specific antibodies and viral-specific CD8+ T cells contribute to vaccination-induced protection.
- Why there are two different figure legends for each figure? This is very confusing and very difficult to follow.
Response: Thank you for pointing this out. We have corrected the mistake.
- All fonts in Figures are of very low resolution, so it is difficult to read and understand some of them.
Response: Thank you. High-resolution pictures are incorporated into the resubmitted manuscript.

Reviewer 2 Report
This manuscript attempted to evaluate new vaccine for Zika virus using a novel single-cycle arbovirus vaccine, in mouse model. This is very important study and the manuscript is well written. After going over manuscript, I would like to give some comments to improve the manuscript.
- Figure legends should be clearly written once instead of having two different descriptions in different areas.
- Text/fonts in the figures are not readable which should be fixed.
Author Response
We would like to thank you for your careful and thorough reading of this manuscript and the thoughtful comments and constructive suggestions, which help improve this manuscript's quality. Our response follows (the reviewer's comments are in italics)
Comments: This manuscript attempted to evaluate new vaccine for Zika virus using a novel single-cycle arbovirus vaccine, in mouse model. This is very important study and the manuscript is well written. After going over manuscript, I would like to give some comments to improve the manuscript.
- Figure legends should be clearly written once instead of having two different descriptions in different areas.
- Text/fonts in the figures are not readable which should be fixed.
Response: We appreciate the positive feedback from you, and thank you for pointing out the shortcomings. We have made changes accordingly: 1. The figure legends have been rewritten. The legends are consistent in both the Figure Legend section and in the body of the manuscript. 2. Higher resolution figures are used in the resubmission.
Reviewer 3 Report
Authors have designed a study to evaluate immunogenicity of VSRD-ZIKV in mouse model by challenging the VSRD-ZIKV immunized mice against two different ZIKA virus strains.
This study has demonstrated robust protection against Asian Zika strain in VSRD-ZIKV immunized mice, humoral and cellular immune responses, protection of male testes and sperm from lethal zika infection and protection against vertical transmission in pregnant mice.
My only issue with this manuscript is lack of WT ZIKA-MR766 challenge on immunized mice for examining robust protection, protection of male testes and vertical transmission in pregnant female mice in sections 3.1, 3.3 and 3.4, respectively. It would have been nice if VSRD-ZIKV immunized mice were examined against both strains.
In addition, introduction section needs to be improved, as it looks a lot like the abstract given the repeated use of several sentences. Moreover, the introduction in this manuscript matches a lot with the first paragraph of their previously published paper, cited in refence 25, thus, it’s better to reorganize.
Overall, I find this manuscript sound, well conceptualized and organized. It would add valuable information on ZIKA vaccine development and would be an interesting read. However, following minor concerns should also be addressed.
- All figures 1-4 are illegible/incomprehensible because of bad image quality, please provide better quality images and modified figure legends directly pasted under each image.
- Figure 1, it will be better to mention the respective ZIKA virus strains used to challenge the sham and immunized group in figure 1C and 1H. In its present form both figures 1C and H cannot be differentiated. And, is it paresis or paralysis in the figure 1C and H texts?
- Figure 4 legend, “Schema” needs to be changed to “Scheme”
- Reference 23 needs a fix.
Author Response
We want to thank you for your careful and thorough reading of this manuscript and the thoughtful comments and constructive suggestions, which help improve this manuscript's quality. Our response follows (the reviewer's comments are in italics)
Comment: Authors have designed a study to evaluate immunogenicity of VSRD-ZIKV in mouse model by challenging the VSRD-ZIKV immunized mice against two different ZIKA virus strains.
This study has demonstrated robust protection against Asian Zika strain in VSRD-ZIKV immunized mice, humoral and cellular immune responses, protection of male testes and sperm from lethal zika infection, and protection against vertical transmission in pregnant mice.
Response: Thank you very much. We appreciate the positive feedback from you.
Comment: My only issue with this manuscript is lack of WT ZIKA-MR766 challenge on immunized mice for examining robust protection, protection of male testes and vertical transmission in pregnant female mice in sections 3.1, 3.3 and 3.4, respectively. It would have been nice if VSRD-ZIKV immunized mice were examined against both strains.
Response: Thank you for the thoughtful comment. We agree that an effective ZIKV vaccine candidate should protect an animal against ZIKV infection of both lineages of ZIKVs circulating in the world. We used WT ZIKA-MR766 challenging the VSRD-ZIKV-immunized AG129 mice in the early stage of this investigation and showed robust protection. Since the VSRD-ZIKV was derived from WT ZIKV-MR766, but serious diseases are often associated with ZIKV of Asian linage, we focused the study on testing if ZIKV-MR766-derived VSRD-ZIKV can induce a strong enough immune response to provide protection against a more lethal and heterologous strain of ZIKV.
Comment: In addition, introduction section needs to be improved, as it looks a lot like the abstract given the repeated use of several sentences. Moreover, the introduction in this manuscript matches a lot with the first paragraph of their previously published paper, cited in refence 25, thus, it's better to reorganize.
Response: Thank you for pointing out the shortcoming. We have rewritten and reorganized the abstract and introduction in the resubmission.
Comments: Overall, I find this manuscript sound, well conceptualized and organized. It would add valuable information on ZIKA vaccine development and would be an interesting read. However, following minor concerns should also be addressed.
- All figures 1-4 are illegible/incomprehensible because of bad image quality, please provide better quality images and modified figure legends directly pasted under each image.
- Figure 1, it will be better to mention the respective ZIKA virus strains used to challenge the sham and immunized group in figure 1C and 1H. In its present form both figures 1C and H cannot be differentiated. And, is it paresis or paralysis in the figure 1C and H texts?
- Figure 4 legend, "Schema" needs to be changed to "Scheme"
- Reference 23 needs a fix.
Response: We appreciate the positive comments from you and thank you for reminding us how important it is to present complex material in high-quality figures:
- Thank you. High-resolution pictures are used in the resubmission.
- Thank you. The ZIKA virus strains are clarified in Figure 1, and the misspelling work "paresis" is changed to "paralysis."
- Thank you. The misspelling word "Schema" is changed to "Scheme."
- Thank you. Reference 23 has been fixed.

Round 2
Reviewer 3 Report
I thank authors for adding requested revisions and improving the overall quality of the manuscript.
All concerns are adequately addressed and the manuscript looks good in its present form.